# An experimental evaluation of the effect of escape gaps on the quantity, diversity, and size of fish caught in traps in Montserrat

Jason Flower[1,2,3]*, Andy Estep[4], Keinan James[5], Robin Ramdeen[4], Claire A. Runge[6], Lennon Thomas[1,2,3], Sarah E. Lester[7]

1 Marine Science Institute, University of California, Santa Barbara, Santa Barbara, CA, United States of America, 2 Bren School of Environmental Science & Management, University of California, Santa Barbara, Santa Barbara, CA, United States of America, 3 Environmental Market Solutions Lab, University of California, Santa Barbara, Santa Barbara, CA, United States of America, 4 Waitt Institute, La Jolla, CA, United States of America, 5 Youth Apprenticeship Program, Ministry of Education, Youth Affairs & Sports, Brades, Montserrat, 6 School of Earth and Environmental Sciences, University of Queensland, St Lucia, QLD, Australia, 7 Department of Geography, Florida State University, Tallahassee, FL, United States of America

* jflower@ucsb.edu

**Data Availability Statement:** All data and R code required to repeat the analyses in this paper are available on Github: https://github.com/emlab-

## Abstract

Coral reef fisheries are vital to the livelihoods of millions of people worldwide but are challenging to manage due to the high diversity of fish species that are harvested and the multiple types of fishing gear that are used. Fish traps are a commonly used gear in reef fisheries in the Caribbean and other regions, but they have poor selectivity and frequently capture juvenile fish, impacting the sustainability of the fishery. One option for managing trap fisheries is the addition of escape gaps, which allow small fish to escape. We compared catches of traps with and without two 2.5 cm (1 inch) escape gaps on the Caribbean island of Montserrat. No significant differences were found in the mean fish length, total fish biomass, number of fish, fish species richness, and Shannon diversity index between hauls of the two trap designs, though traps with escape gaps did catch larger proportions of wider-bodied fish and smaller proportions of narrow-bodied fish. Furthermore, traps with gaps caught a smaller proportion of small-sized fish and fewer immature fish (though differences were not statistically significant). Linear mixed effect models predict that soak time (the length of time between trap hauls) increases the mean catch length, total catch biomass and total number of species in the catch. The relatively modest evidence for the effect of the gaps on catch may be explained by the long soak times used, which could have allowed most smaller-sized fish to escape or be consumed by larger individuals before hauling in both traps with and without escape gaps. Despite the small differences detected in this study, escape gaps may still offer one of the best options for improving sustainability of catches from fish traps, but larger escape gaps should be tested with varying soak times to determine optimum escape gap size.

ucsb/montserrat-trap-experiment-paper and
Zenodo: http://doi.org/10.5281/zenodo.5139291.

**Funding:** This project was funded by the Waitt
Foundation, https://www.waittfoundation.org/. The
Waitt Institute provided staff and resources to
conduct the study, but they played no role in the
study design, analysis, or decision to publish.

**Competing interests:** The authors have declared
that no competing interests exist.

## Introduction

Reef fisheries provide livelihoods for an estimated 6 million reef fishers worldwide [1], and
reef fish provide an important source of income and nutrition to some of the world's poorest
people [2, 3]. Many reef fisheries are overfished [4] and the degradation of reef habitats due to
global climate change and local human impacts is further reducing reef fisheries productivity
[5, 6]. Effective management of reef fisheries is particularly challenging due to their multi-
gear, multi-species nature. Spatial management, such as the use of marine protected areas
(MPAs), is commonly used as a fisheries management tool [7, 8], but MPAs are often opposed
by fishers [9, 10] and can be challenging to enforce. Gear-based management, where fishing
gear is regulated to control selectivity, or the timing and location of gear deployment is con-
trolled, can be more acceptable to fishers and lower cost to implement [9, 11].

Fish traps are one type of fishing gear that is commonly used throughout the Caribbean,
West Pacific and Western Indian Ocean [12]. Traps are popular with reef fishers as they are
relatively simple and inexpensive to make [13], can be used on rugged substrates where other
gears might be damaged [14], and they catch fish without the fisher having to attend them and,
as such, can be left at sea in inclement weather [15]. However, they present a challenge for reef
managers as they can negatively impact fish stocks and ecosystems in a number of ways. First,
the small mesh sizes that are frequently used in traps result in the capture of juvenile fishes
[16]. Second, traps have poor selectivity–they will retain any fish that enters the trap and is not
able to find its way back out of the entrance funnel or escape through the mesh. This indis-
criminate harvest results in high catch diversity, the capture of many species that are not of
commercial interest, and the inability to avoid catching species that are known to be over-
fished. Fish traps catch herbivorous fish, including parrotfish (Scaridae) and surgeonfish
(Acanthuridae), which play a vital role in facilitating coral growth and recovery by controlling
the abundance of macroalgae on reefs [17, 18]. Third, traps themselves can cause direct physi-
cal damage to habitats, such as scarring or breaking corals [19]. Finally, traps are frequently
lost or abandoned. These traps will continue to catch fish for as long as their physical structure
is intact, called "ghost fishing", contributing to fish mortality and marine debris.

To mitigate some of the problems posed by fish traps, a number of options are either in use
or have been trialled: varying the size of the mesh used on the trap [e.g. 20–22], adding escape
gaps to traps to allow some fish to swim out [e.g. 23, 24], adding biodegradable escape panels
or panel closures to traps to minimize ghost fishing [25], and retrieving derelict (ghost) gear
[e.g. 26, 27]. Increasing mesh size and adding escape gaps can both reduce the number of juve-
nile (immature) fish caught by traps, with escape gaps also allowing for the escape of narrow-
bodied fish that can have little commercial value but are functionally important [23].

Compared to mesh size, relatively little research has been published on the effects of escape
gaps on trap catches [23, 28]. Escape gaps are narrow slits that are built into a trap, with tested
sizes varying from 2 to 8 cm in width and 3 to 40 cm in height (Table 1). Previous studies have
found escape gaps can reduce the catch of juvenile fish and increase the mean length of landed
fish [23, 24, 29, 30]. Furthermore, some studies have found that even when the mean weight of
the catch is reduced, there can be minimal impacts to the market value of the catch due to
retention of larger, higher value fish [23, 24, 29]. Because smaller and narrow bodied species
can escape, the species composition of the catch can change, in some cases reducing the num-
bers of functionally important fish, such as herbivores, that are caught [23, 30]. There have
been only two studies in the Caribbean exclusively examining the impact of escape gaps on
catch [23, 28], despite the widespread use of fish traps in the region [16, 31].

To further the understanding of the impact of escape gaps and other factors on fish trap
catches, we conducted experimental trials of traps with (experimental) and without (control)

**Table 1. Summary of results on fisheries catches of all experimental studies of fish traps with escape gaps for multi-species coral reef fisheries.**

| Study | Location | Trap configurations tested | Soak time (days) | Change in catch per trap haul in traps with escape gaps relative to traps without escape gaps | | | | Further information |
|---|---|---|---|---|---|---|---|---|
| | | | | Mean length | Mean catch biomass (except where indicted) | Mean number of fish | Mean value of catch | |
| Condy et al. 2015 | Kenya, East Africa | 1 gap, widths 2, 4, 6 and 8 cm | 1 | ⬆ All mean lengths greater than control, greatest length in 4 cm gap traps | ⬆ Mean fish weight per haul reported. All means greater than control, greatest increase in 4 cm gap traps | ⬇ Decreased with increasing gap width | ⬇ Decreased with increasing gap width | 2 cm escape gap suggested as best option for reducing catch of juveniles and functionally important algal browsers, while minimising economic impact on fishers |
| Gomes et al. 2014 | Kenya, East Africa | 2 gaps, 3 cm wide | 1 | ⬆ (only 1 species tested) | ⬆⬇ Decreased in low exploitation areas, slight increase in overexploited areas (*NS) | | ⬆⬇ Decreased in low exploitation areas (*NS), increased in overexploited areas | Traps with escape gaps caught less low-value fish (juveniles and narrow-bodied species) |
| Johnson 2010 | Curacao, Caribbean | 2 gaps, 2.5 cm width, lengths 20 and 40 cm | 1 | ⬆ Increases with gap length | ⬇ *NS in tall gap | ⬇ *NS in tall gap | ⬇*NS | Reduced bycatch and catch of herbivores in traps with escape gaps |
| Mbaru & McClanahan 2013 | Kenya, East Africa | 2 gaps, 4 cm x 30 cm | 1 | ⬆ | ⬆ mean fish weight per haul | - | ⬆ | Bycatch of butterflyfish and other low value species lower in traps with escape gaps |
| McClanahan & Kosgei 2018 | Kenya, East Africa | 2 gaps, 2–4 cm gaps, | Not stated | ⬆ | | | - | Outcomes influenced by competition with other fishers using other gears |
| Munro et al. 2003 | Jamaica and British Virgin Islands, Caribbean | 2 gaps, widths 2.5–3.3 cm, lengths 7–9 cm | 3–7 | ⬇ Decreased with gaps size | | ⬇ Decrease with gap size | | Introduction of smallest gaps and gradual increase in gap size predicted to increase fishery yields while minimising impact on fishers |

⬆ indicates an increase relative to traps without escape gaps, ⬇ indicates a decrease, ⬆⬇ indicates mixed effects, and—indicates no change.

*NS = Not statistically different (p>0.05).

two 2.5 cm (1 inch) escape gaps on the Caribbean island of Montserrat, where the trap fishery includes both coral reef and demersal species. Based on past evidence, we hypothesized that:

1.  Mean length and weight of fish in experimental traps will be larger than control traps

2.  Total catch biomass will be greater in control traps than in experimental traps

3.  The number of fish caught in control traps will be greater than in experimental traps

4.  Species composition of fish will be different in control and experimental traps

5.  The numbers of juvenile fish caught in experimental traps would be lower than in control traps

Our experimental design was similar to previous experiments, with traps deployed in pairs, but we also tested two trap designs, set trap pairs at differing depths, and had variable soak times (time between hauls when the catch is removed) over the length of the experiment. This allowed us to explore the impacts of these factors on catch for traps with and without escape gaps.

## Methods

### Site description

Montserrat is a volcanic island in the north-eastern Caribbean, with a population of 4,490 (pers. comm. Statistics Department of Montserrat, 21 March 2019), and a total land area of 102 km² and 49 km of coastline. It has a narrow shelf, covering approximately 141 km² out to the 100m depth contour, with most of the shelf area around the northern half of the island, while the southern end of the island, around the volcano, drops off steeply (Fig 1). The shelf seafloor is dominated by sand and algal reefs, with coral reefs found along the east and north-eastern sides of the island, and a thin strip of reef along the southeast coast [32]. Access to the southern half of the island is restricted due to the active volcano, and maritime access to the surrounding coastal waters is limited (Fig 1; www.mvo.ms).

Montserrat's artisanal fishery targets coral reef, demersal, coastal pelagic, and pelagic species and in 2015 there were 34 active fishers and 29 active fishing vessels [33]. All catches are

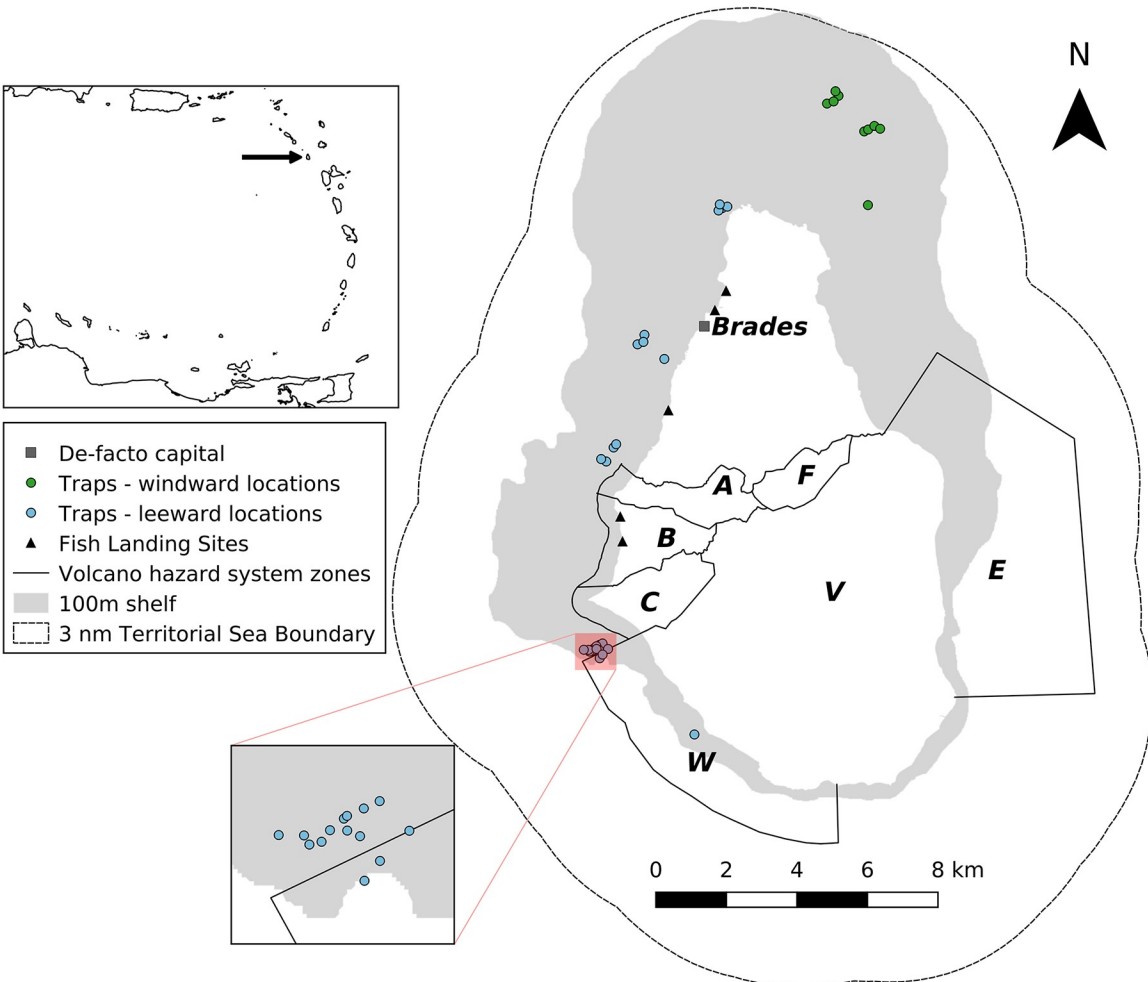

**Fig 1. Map of Montserrat showing locations of fish traps.** Inset map shows location of Montserrat (indicated by arrow) in Lesser Antilles island chain. At hazard level 1 (as of 9 July 2021) zones A, B, C, and F of the volcano hazard system allow unrestricted access, zone V is controlled access, and maritime zones E and W allow daytime transit only. For current hazard level see: www.mvo.ms. Map created using QGIS software with Natural Earth (public domain) data for the inset: http://www.naturalearthdata.com/, and data from the Government of Montserrat and this study for the main map.

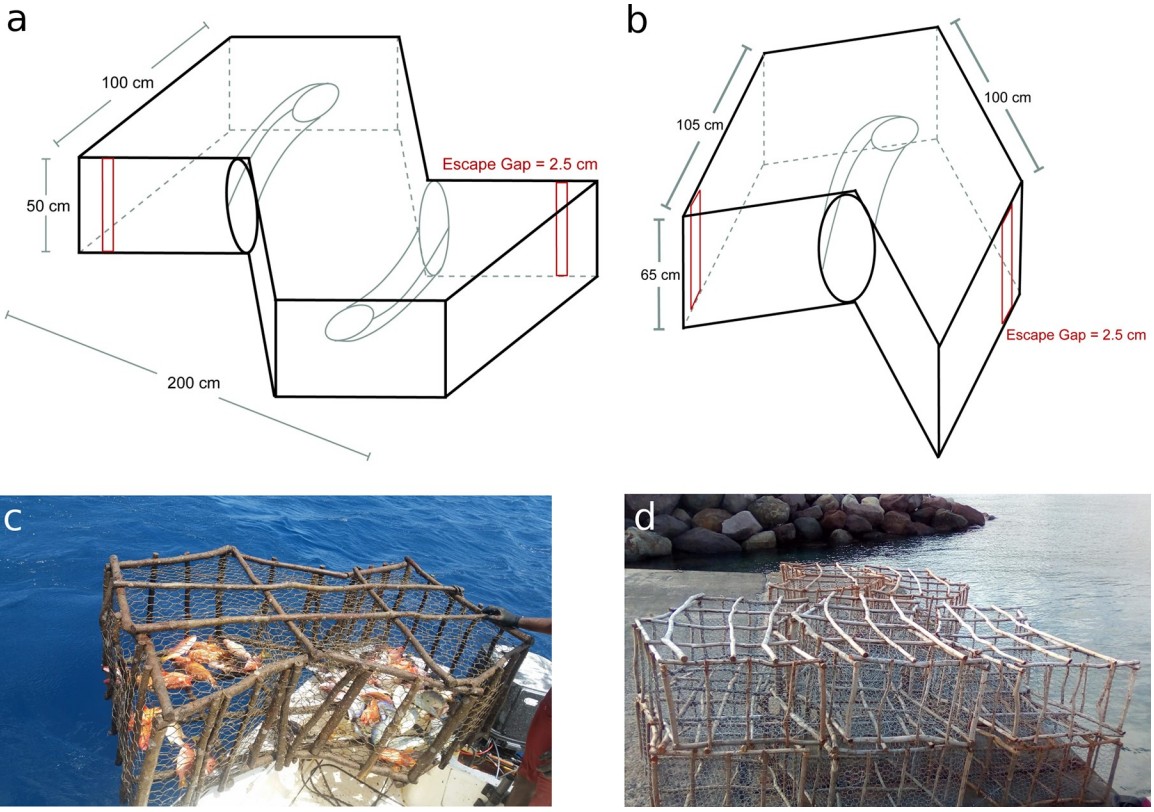

**Fig 2. Diagrams and photos of traps used in the experiment.** (a) schematic of Z-trap showing dimensions and locations of gaps and funnels; (b) schematic of chevron trap showing dimensions and locations of gaps and funnel; (c) photo of Z-trap with catch in; (d) photo of chevron traps on shore.

for subsistence or for sale on the local market; there is no export market [34]. The local market for fish operates on a flat rate price per pound (weight) of unprepared fish, which was EC$10 (= US$3.70) as of November 2019. The majority of reported catch is landed at five landing sites on the western side of the island (Fig 1). Fish traps account for approximately half of total recorded landings [33] and their catch is made up of coral reef and demersal species. Traps are made of a wood frame covered in 3.8 cm (1.5 inch) wire mesh. Two types of trap designs are used: chevron (or V) traps and Z traps (Fig 2). Traps are used with and without bait, and left to soak (time between hauls where catch is removed) for a minimum of 3 days, and several weeks if sea conditions are not favorable to retrieve traps. Traps are only brought back to land if they are in need of repair. All species of caught fish are landed, but smaller individuals are discarded at sea when traps are hauled (pers. obs. L.T., 19 June 2015). In 2015, 157 traps were observed in-water, all within 4.8 km (3 miles) from shore and between 15 and 100 m depth [33].

## Experimental design

To test the impact of escape gaps on catch and catch composition, we deployed a total of 40 traps between March and November 2018 in experimental-control trap pairs (one trap with and one without escape gaps) placed in close proximity to each other (Fig 1). Each experimental trap had two escape gaps, approximately 2.5 cm (1 inch) in width (exact width varied between 1.9 and 3.8 cm because unfinished wood was used to make the gaps), that extended the full height of the trap, but were otherwise identical to control traps (Fig 2). Half of the traps

were V-shaped and the other half were Z-shaped. All traps were made using 1.5 inch (3.8 cm) wire mesh, and fitted with escape panels to minimize ghost fishing impacts if lost. Traps were located in areas known to be used by trap fishers in Montserrat and each experimental-control trap pair (20 pairs in total) was deployed at a different site (Fig 1), with each pair next to each other and individually tethered to a buoy on the surface. Traps were not baited due to inconsistent availability of the bait that some fishermen use (coconut). The positions of traps were kept approximately constant throughout the experiment and were recorded using a handheld GPS. Trap depths were recorded using a depth sounder and were in the range 15.2 to 50.3m. Although trap pairs were deployed at the same location, natural movement of the traps due to ocean conditions, and the steep shelf slope around Montserrat could result in differing depths within a trap pair.

Traps were deployed in batches, with 10 pairs (experimental and control) deployed on 12th March 2018, 5 pairs on the 24th April 2018, and a further 5 pairs on the 14th August 2018 (S1 Fig). The staggered deployment was to allow for an initial test period before deploying all traps and because of limited space on the boat and delays in trap construction. Soak time was initially set at 4 days, but was increased to 1 week, and finally 2 weeks to more closely match realistic soak times used by local fishers and to reduce the frequency of the fishing trips. Hauling of traps was sometimes delayed due to bad weather, and individual traps could not always be hauled on a trip due to ocean conditions and problems locating them. There were a total of 23 days when traps were hauled between 16[th] March 2016 and 21[st] November 2018 (S1 Fig), and 12 traps were lost during the experiment. When hauled, all fish in each trap were put in labelled bags, and species, weight (to the nearest gram) and length (to the nearest 0.5 cm) of each fish was recorded at the landing site.

The experimental fishing and data collection was done by apprentices employed through the Government of Montserrat's Ministry of Education, Youth and Sports, and the Ministry of Agriculture. A single data collector (K.J.) collected all data and received training before the experiment commenced. No permit was required for the field work because the relevant government departments were directly involved in the project. No university researchers were involved in the experimental fishing.

## Analysis

**Catch comparison.** We compared the total number of fish, mean length of fish, total weight of fish (biomass), total number of species (i.e. species richness), and Shannon diversity index per haul in control and experimental traps using only data from trap pairs hauled together. Mean length and weight of fish were also compared using individual fish as the sampling unit. Data were log-transformed for normality when necessary, and the comparisons were made using paired t-tests. Non-parametric, Wilcoxon-signed rank tests were also done on the non-transformed data as a check of the results. Comparisons were made with and without hauls with no fish due to uncertainty as to whether these traps had been poached or were true zero catches.

We compared the proportions of each species caught in paired control and experimental hauls using chi-squared tests.

**Length-based analysis.** We compared the length-frequency distributions of all fish caught in control and experimental traps (paired trap haul data with zero hauls excluded) using Kolmogorov-Smirnov tests. We also compared the distributions of only small-sized classes of fish (<20 cm length and <15 cm length) as we expected the biggest difference between control and experimental traps to be in the catch of narrower, and therefore shorter, fish. As a further investigation of the data, we compared the length-frequency distributions of each species in

the two trap types, limiting the analysis to the 18 species for which more than 10 individuals were recorded in both control and experimental traps.

To see if the experimental traps would reduce the proportion of immature fish caught, we compared the percentage of fish caught below the length at maturity ($L_{mat}$; length at which at least 50% of individuals are mature) in control and experimental traps hauled as pairs. We used $L_{mat}$ values from two sources that had relatively local, reliable information [33, 35], providing $L_{mat}$ values for 9 species, representing 54% of the total catch by number. For these 9 species, we compared the number of fish less than $L_{mat}$ in paired control and experimental traps using paired Wilcoxon-signed rank tests since the data were non-normally distributed.

Due to the importance of soak time as an explanatory variable in the linear mixed effects models (see Results—Linear mixed effects models), we also examined the length-frequency distributions of fish caught in traps (all trap hauls excluding zeros) with soak times classified as short (4, 6, and 7 days), medium (12 and 14 days) and long (29 and 30 days).

**Body width-based analysis.** To examine the effect of the escape gaps on retention of fish according to body width, we classified each species using descriptions of body shape from a curated database of Caribbean fish [36], with some modifications to the classification after communication with the database maintainer (pers. Comms. R. Robertson, 11[th] March 2021). Fish were classified from narrowest (most compressed) to widest (not compressed) with those described as "very compressed" or "strongly compressed" classified as "very compressed", those as "moderately compressed" or "somewhat compressed" classified as "moderately compressed", those as "compressed" classified as "slightly compressed", and those as "thick" or "robust" as "not compressed". Differences in the proportions of fish in each group caught in control and experimental traps were examined using chi-squared tests.

**Linear mixed effects models.** We examined the effect of escape gaps on mean length, total biomass, total number of fish, and species richness of the catch in each trap haul. Four fixed effects were considered in all the models: soak time (days), trap design (V or S shape), location (leeward or windward side of the island), and experimental or control trap (with or without escape gaps). Soak time was log transformed because response to increasing soak time is expected to be asymptotic [16]. As depth was not recorded for all traps, models using a subset of the data with depth were fitted separately. However, depth was not a significant predictor in any model ($p > 0.05$ in global models and not included in most parsimonious models), therefore these results are not reported further. Two random effects were included in all models: the date of the trap haul and trap ID which identifies each individual trap. These random effects were incorporated to control for pseudoreplication and temporal auto-correlation. Separate linear mixed effects models (LMMs) were fitted to the mean length of fish per trap haul and total weight of fish per haul (log+1 transformed to meet conditions of normality). Negative binomial generalized linear mixed effects models (GLMMs), were fitted to total number of fish per haul and number of species of fish per haul. Negative binomial models were chosen to account for positively skewed count data, and were preferred over poisson models because the data are overdispersed and negative binomial models perform better in this case [37]. The LMMs were fitted using the 'lmer' function and GLMMs were fitted using the 'glmer.nb' function in R [38], both from the 'lme4' package [39]. For the LMMs a backward selection algorithm was employed using the 'lmerTest' package [40] 'step' function to find the most parsimonious model with the greatest explanatory power. For the GLMMs, we used AIC to compare model fits, with lowest AIC selected as the most parsimonious models. We used the 'dwplot' function from the 'dotwhisker' package [41] to plot regression coefficients and compare all potential models using all combinations of fixed effects. Model diagnostic plots, summary tables, and prediction plots were produced using the 'sjPlot' package [42].

## Results

### Catch characterization and comparison

A total of 23 days at sea were spent hauling traps, and fish from 21 families and 58 species were recorded. Traps were hauled a total of 336 times; 157 hauls of control traps and 179 hauls of experimental traps. A total of 135 of these hauls were paired traps, with the remaining 22 control traps and 44 experimental traps hauled singly due to the paired trap being lost or unable to be located on that day. Twelve hauls of the control traps and 17 of the experimental traps had no fish (Table 2).

Looking at all catch data combined, doctorfish (*Acanthurus chirurgus*) and blue tang (*A. coeruleus*) were the most common species in the catch, representing 20.3% and 14.5% of all fish by number, respectively (Fig 3). Whitespotted filefish (*Cantherhines macrocerus*; 11.9%), honeycomb cowfish (*Acanthostracion polygonius*; 8.4%), red hind (*Epinephelus guttatus*; 5.1%), squirrelfish (*Holocentrus adscensionis*, 4.0%), and French grunt (*Haemulon flavolineatum*; 4.0%) were the next most common species, with the remaining 48 species each representing less than 3% of the total number of fish, and combined totalling 31% of the catch by number.

No significant differences were found in number, length, biomass, Shannon diversity index, and species richness of fish in control versus experimental traps when grouped at the haul level, and no significant differences were found in length and weight of fish when grouped at the individual level (S2 Table; p> 0.1 in all cases).

Only nine species were caught in significantly different proportions in control and experimental traps (S3 Table). Most of these represented very small proportions of the catch (<2% in either trap), but black durgon (*Melichthys niger*), which is a narrow-bodied fish, was absent from experimental traps, but formed 2.1% of the total catch in control traps ($\chi^2(1) = 24.2$, p < 0.001). Spotfin butterflyfish, another narrow-bodied species, was only 3.0% of the control catch compared to 1.4% of the experimental trap catch ($\chi^2(1) = 7.0$, p = 0.008). Notably, coney (*Cephalopholis fulva*) and Spanish hogfish (*Bodianus rufus*), preferred food fish in the Caribbean, were caught in higher proportions in the experimental trap ($\chi^2(1) = 3.9$, p = 0.049 and $\chi^2(1) = 11.5$, p = 0.001 respectively), although this result is based on relatively small sample sizes (S3 Table).

**Table 2. Summary of environmental and catch data for control traps (without escape gaps) and experimental traps (with escape gaps).**

|  | Only data from paired trap hauls | | All traps | |
|---|---|---|---|---|
|  | Control | Experimental | Control | Experimental |
| No. of hauls | 135 | 135 | 157 | 179 |
| No. of hauls with zero fish | 11 | 16 | 12 | 17 |
| No. individual fish measurements | 1320 | 1253 | 1503 | 1707 |
| Total no. species | 52 | 43 | 52 | 49 |
| Mean water depth of traps (m) | 36.8 ± 12.9 | 32.5 ± 10.4 | 36.8 ± 12.9 | 32.5 ± 10.4 |
| Mean trap soak time (days) | 11.9 ± 7.6 | 11.9 ± 7.6 | 11.6 ± 7.5 | 11.6 ± 7.6 |
| Mean number of fish per haul | 10.8 ± 9.1 | 10.4 ± 9.0 | 10.4 ± 9.2 | 10.5 ± 10.0 |
| Mean length of fish per haul (cm) | 25.2 ± 4.7 | 25.4 ± 4.7 | 25.2 ± 4.6 | 25.2 ± 4.3 |
| Mean total biomass of fish per haul (kg) | 4.0 ± 3.7 | 3.7 ± 3.4 | 3.7 ± 3.5 | 3.7 ± 3.3 |
| Mean number of species per haul | 4.0 ± 2.3 | 4.0 ± 2.4 | 3.9 ± 2.3 | 3.9 ± 2.5 |

Data for all traps and only trap pairs hauled together are presented. All means are presented ± standard deviation, and exclude data from traps with zero fish. An expanded version of this table is presented in S1 Table.

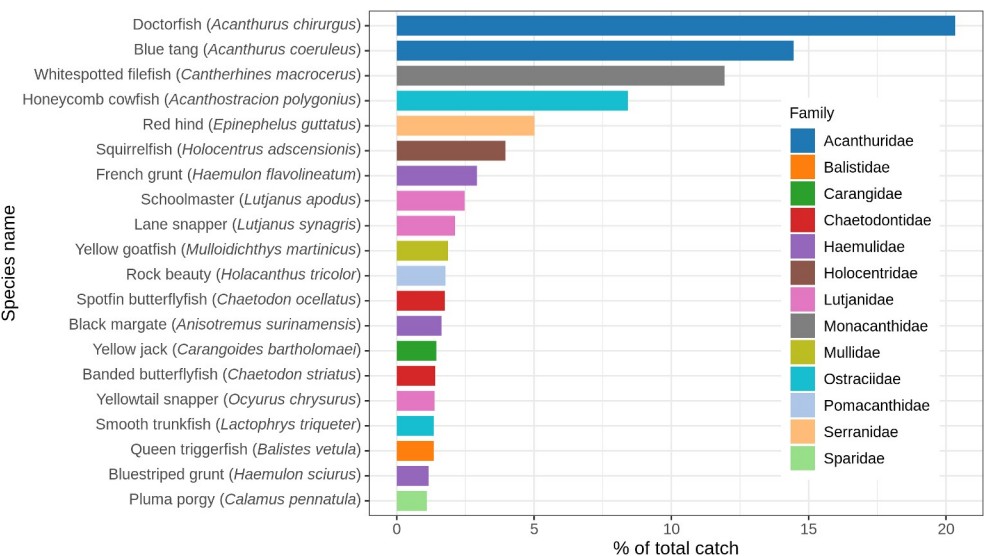

**Fig 3. Species composition (number of individuals) of all catches combined.** Species comprising less than 1% of the catch by number are not shown.

## Length-based analysis

There was no significant difference in the length-frequency distributions of all fish from control and experimental traps (using paired trap haul data only; Kolmogorov-Smirnov $D_{max}$ = 0.02, p = 0.94; Fig 4). Comparing only small size classes of fish (i.e. all fish less than a threshold size) revealed no significant difference in length-frequency distributions for both fish in size classes less than 20 cm (Kolmogorov-Smirnov $D_{max}$ = 0.07, p = 0.36) and those less than 15 cm (Kolmogorov-Smirnov $D_{max}$ = 0.11, p = 0.97). Although not statistically significant, there were a greater proportion of fish less than 15 cm in length in the control traps (6.3% of total number of individuals) compared to the experimental trap (4.1%), with the difference principally being driven by the greater number of spotfin butterflyfish (*Chaetodon ocellatus*) caught in the control traps (36 individuals in control, 16 in experimental), and small size blue tang (S2 Fig).

Out of 18 species analysed, none had significant differences in their length frequency distributions between control and experimental traps (Kolmogorov-Smirnov test p > 0.05 in all cases; S4 Table).

Of the nine species for which we had length at maturity values, 5 had a higher percentage of immature fish (more fish less than $L_{mat}$) caught in the control traps than the experimental traps, 1 species had a higher percentage in the experimental traps, and the remaining 3 were all caught above $L_{mat}$ in both trap types, though none of the differences were statistically significant (p > 0.1 in all cases; S5 Table).

Traps left for longer soak times appear to shift to larger size distributions (S3 Fig), with the difference being driven by a reduction in small sized fish (< 20 cm), principally composed of blue tang and doctorfish, to fish in larger size classes, including honeycomb cowfish, red hind, and schoolmaster snapper (*Lutjanus apodus*).

## Body width-based analysis

The proportion of very compressed (narrowest) fish found in control traps was significantly higher than that in experimental traps ($\chi^2(1)$ = 4.3, p = 0.04, Fig 5). Conversely, the proportion

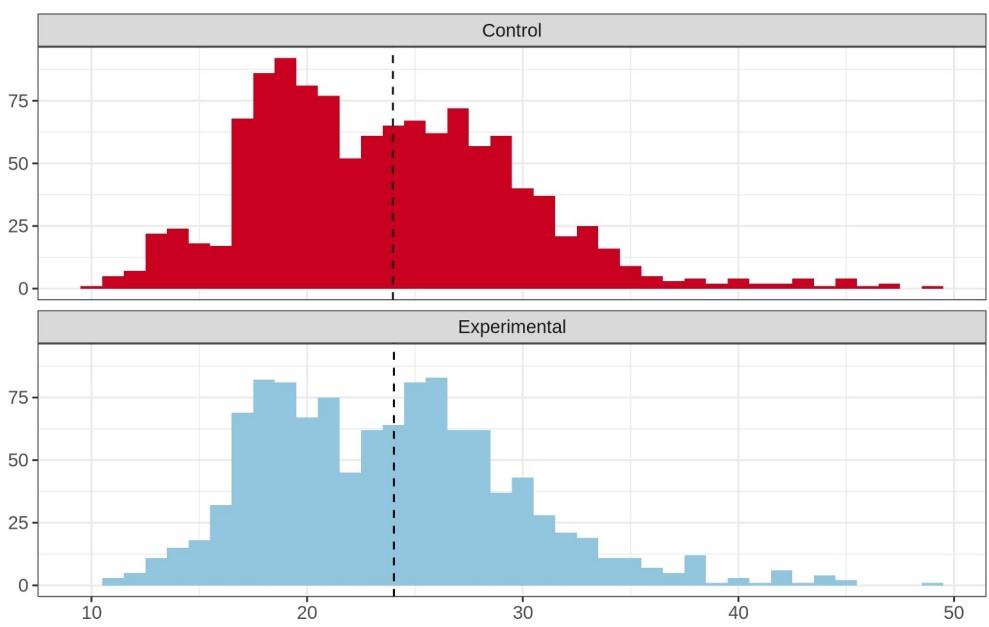

**Fig 4.** Length-frequency distributions for fish in control (top) and experimental (bottom) traps, using data from paired trap hauls, excluding hauls with zero catch. Dashed vertical lines show mean length of fish. Fish greater than 50 cm (n = 6) have been omitted for clearer visualization.

of not compressed (widest) fish was significantly higher in experimental traps than control traps ($\chi^2(1)$ = 4.6, p = 0.03). The proportions of fish classified as moderately and slightly compressed were not significantly different between control and experimental traps.

## Linear mixed effects models

Of the four fixed effects considered in all models, only soak time and location were significant predictors (Table 3 and S6 Table). In the global models (all fixed effects included), soak time and location were significant predictors of mean catch length (p < 0.001 and p = 0.018 respectively) and soak time was the only significant predictor of total catch biomass (p < 0.001; S6 and S7 Tables).

The most parsimonious length model only included soak time as a predictor, with mean catch length increasing asymptotically from 22.9 cm (95% Confidence Interval 21.8–24.0 cm) for a 4 day soak time to 28.1 cm (26.7 cm—29.4 cm) for a 30 day soak time (Table 3 and Fig 6). The most parsimonious biomass model also included only soak time as a predictor, with total catch biomass increasing from 1.8 kg (1.3–2.4 kg) to 4.2 kg (3.1–5.6 kg) for 4 and 30 day soak times respectively (Table 3, Fig 6). Location was a predictor in the most parsimonious number of fish and species richness models, though was not significant (p = 0.08 and p = 0.11 respectively), and soak time was also a significant predictor in the species richness model (p = 0.04). Traps in windward locations have a trend of catching fewer fish (6.9, 95% CI 4.8–10.2) than leeward locations (9.9, 95% CI 8.3–11.8; Fig 6), and also catch fewer species of fish (3.1. vs. 4.0 after 14 day soak time, 95% CIs: 2.4–4.1; 3.5–4.5; Fig 6).

All model estimates have large confidence intervals, in part due to the large variance in the data, and in the case of the effect of location, due to the small sample size for traps in windward locations (n = 30) compared to leeward locations (n = 277). Low $R^2$ values for all models (<0.3 in all cases, S7 and S8 Tables), indicate they explain a relatively low proportion of the total variance.

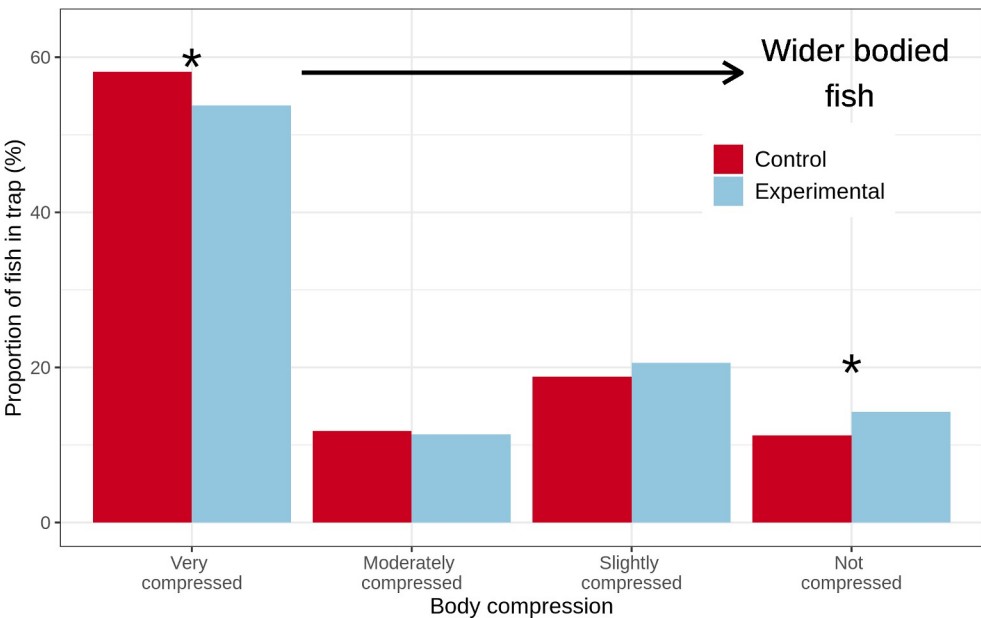

**Fig 5. Proportion of fish in control and experimental traps classified by body width.** Asterisk above bars indicates there was a significant difference between control and experimental traps ($p < 0.05$, chi-squared test).

## Discussion

We found no significant effect of escape gaps on mean fish length, total fish biomass, number of fish, Shannon diversity index, and species richness per trap haul. Despite a lack of differences in these measures of catch, more detailed analysis of the data did reveal effects of escape gaps on catch. Specifically, the narrowest fish were found in significantly higher proportions in control traps compared to experimental traps and conversely the widest fish were found in higher proportions in experimental traps than control traps. Additionally, black durgon (a narrow-bodied triggerfish) were absent in the experimental traps but formed 2.1% of all fish in control traps and there was a greater proportion of small-sized (less than 15 cm length) fish in the control traps (though not significant). Furthermore, of the 9 species for which we had length at maturity values, 5 had a higher percentage of immature fish caught by the control trap compared to the experimental traps, with only 1 species having a higher percentage in the experimental trap, though these results were not statistically significant due to the relatively

**Table 3. Fixed effects present in the most parsimonious model and ΔAIC between most parsimonious model and global model.**

|  | **Mean Catch Length** | **Total Catch Biomass (log+1)** | **No. of fish in catch** | **Species richness** |
|---|---|---|---|---|
| Fixed effects in most parsimonious model | (log) Soak time** | (log) Soak time** | Location - | (log) Soak time* |
|  |  |  |  | Location - |
| ΔAIC most parsimonious: global model | 0.4 | 3.9 | 4.6 | 3.2 |

Most parsimonious models were selected by backwards reduced term elimination for length and biomass models, and lowest AIC for number of fish and species models. ΔAIC < 2 indicates models are indistinguishable and ΔAIC < 6 is commonly used as a cut-off to find the 'best' models [43].—p > 0.05

* p < 0.05

** p < 0.001.

See S8 Table for model summary.

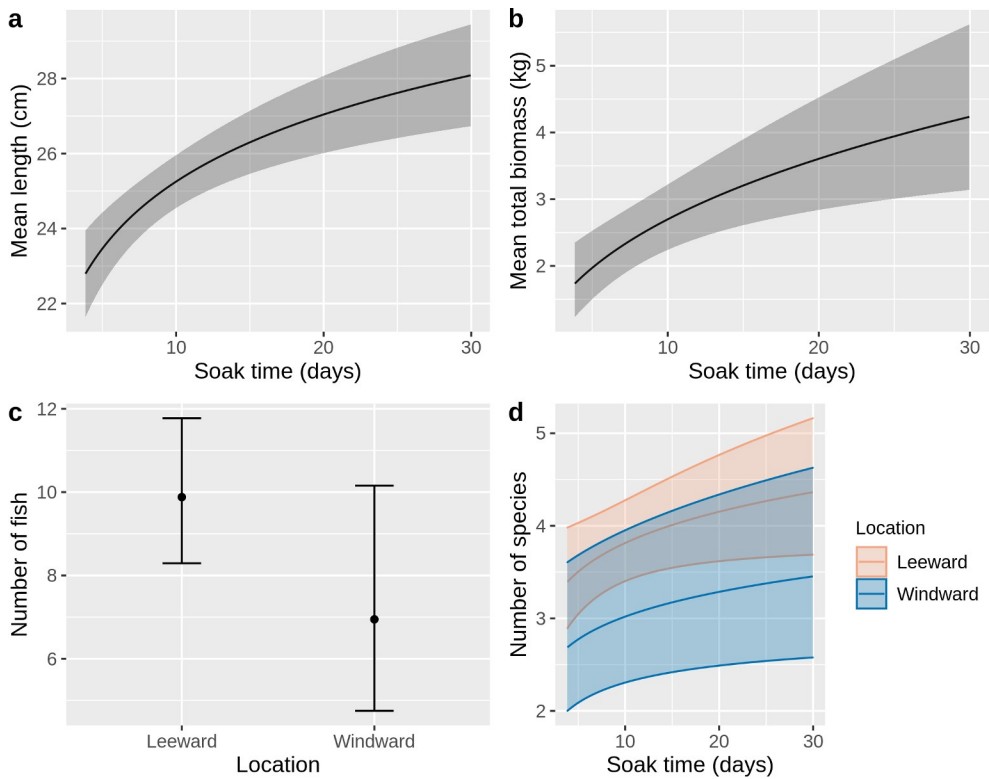

**Fig 6. Marginal effects plots for most parsimonious models.** Describing variation in: (a) mean length of fish in haul; (b) mean total biomass of fish in haul; (c) number of fish in haul; (d) species richness.

small magnitude of the differences. Taken together, these results suggest the escape gaps allowed some narrow bodied and immature fish to escape.

The similarity of some measures of catch per trap haul (length, biomass, number of fish, species diversity and richness) in control and experimental traps is in contrast to previous experiments in coral reef fisheries that have generally found that traps with gaps catch less fish, the mean length of the catch is greater, and the total biomass of fish caught is different (Table 1), though it should be noted our traps included both coral reef and demersal species. One possible explanation for the limited difference between our control and experimental trap catch is that the longer soak times of our traps resulted in most smaller fish exiting or being consumed in both trap types before hauling. The mean soak time for all traps was 11 days (range 4–30), which is much higher than all past experiments, which have mostly had 1 day soak times (Table 1), but is consistent with fishing practices in Montserrat. The longer soak times may increase the number of larger, predatory fish, which are only attracted to the traps once there is a minimum abundance of prey species. Once these larger fish have entered, the exit rate of smaller species, which are known to have high escapement rates from traps through the funnel, may increase, or they may be consumed, leaving higher proportions of larger species which are less able to escape [44]. This explanation is consistent with our results which show a shift to larger sized fish in the catch from all trap types with increasing soak time (S3 Fig), and is further corroborated by the fact that the mean length of fish caught in our trap hauls (24 cm) was larger than all comparable studies [23, 24, 30, 45].

Other potential reasons that our escape gaps did not have a significant impact on most measures of catch are that fish were not able to find the gap or that smaller sized fish were able to

exit traps through the mesh or entry funnel of control traps. The escape gaps we used were made of the same material as the rest of the trap whereas some past studies have used metal rebar to make the gaps [23, 24], which could make them more visible to fish. However, another study that used similar methods to ours for creating the gaps found a significant effect on catch [30], suggesting that visibility of the gap may not have been a factor. The exit of smaller sized fish through the mesh and funnel is a more feasible explanation as fish are known to pass in and out of traps relatively freely, with one study finding almost equal numbers of fish entering and exiting the trap [44] and another finding that 51% of black sea bass entering the trap exited before hauling [46]. The relatively free movement of smaller sized fish in and out of all traps would explain the limited difference between experimental and control traps, but not the shift to larger sized fish with increasing soak time.

Soak time was a significant predictor of mean fish length, total fish biomass, and species richness in the catch, with longer soak times increasing all values towards an asymptote. Previous studies of the impact of soak time on catches in coral reef fisheries found a levelling-off or decline after 4 to 7 days soak time as entry and exit rates of fish converge [16], however, this is in part an artifact of the maximum soak times in those studies, because experiments with longer soak times (14–20 days) found the number of fish caught was still increasing albeit more slowly [47, 48]. Our results contrast with this, showing no significant change in number of fish with soak time, but increasing total biomass and mean length of fish caught, i.e., a shift to larger sized fish with increasing soak time. As explained previously, this may be due to larger fish entering the traps to prey on smaller fish.

We found a trend (though not significant) of more fish and more fish species caught in traps placed on the leeward (west) side of the island compared to those on the windward side. This fits with the ecology of Caribbean islands where wave exposure is known to drive patterns of abundance and species composition [49–51]. It is also consistent with an assessment on Montserrat's coastal waters that showed a general pattern of lower fish biomass and species richness on windward sites [32].

Our findings suggest that, in the case of Montserrat, 2.5 cm (1 inch) escape gaps in fish traps should have minimal impact on fishers' catches and therefore their income. The potential reduction in immature and narrow-bodied fish in the catch could have ecological and fisheries benefits, but given the relatively weak effect of the gaps, larger gaps may be more beneficial, though these should be trialled before being introduced. Our results contrast with previous experiments, and indicate there may only be modest sustainability benefits of escape gaps at longer soak times, though those longer soak times might naturally catch fewer small sized fish. The ecological benefits of longer soak times might be limited if the shift to larger sized fish is driven by consumption of the smaller sized fish by predators, therefore further research is warranted to investigate the mechanisms of changing catch composition with increasing soak times. The higher proportions of two preferred Caribbean food fish (coney and Spanish hogfish) in the experimental traps, is also worthy of further investigation, as this could be a further benefit of escape gaps if the result holds in other locations and it is not clear why these species were more commonly caught in the experimental traps. It is important for managers to note that escape gaps did not have any measured negative impacts on catch, and therefore fitting traps with escape gaps, as well as escape panels to reduce ghost fishing, may offer some of the best options for improving sustainability of catches from fish traps.

## Supporting information

**S1 Fig. Dot plot of dates when each trap was hauled.** Presence of a dot on a date indicates that the trap was hauled, absence indicates it was not hauled. Experimental (E) and Control

(C) traps were deployed in pairs which have the same number, e.g. E1 and C1 are an experimental-control trap pair.
(TIF)

**S2 Fig. Length-frequency distributions for fish in a. control and b. experimental traps, using data from paired trap hauls excluding hauls with zero catch.** Numbers above bars show percentage of total number of fish that each bar represents. Bar fill colour shows species that were more than 2% of the total number of fish (control and experimental traps catch combined), with all other species classified as "other". Fish greater than 50 cm (n = 6) have been omitted for clearer visualization.
(TIF)

**S3 Fig.** Length-frequency distributions of fish species caught in all trap hauls (excluding zero hauls) with: (a) short soak times (4, 6 and 7 days, n = 148); (b) medium soak times (12 and 14 days, n = 124); (c) long soak times (29 and 30 days, number of hauls = 35). Bar fill colour shows species that were more than 2% of the total number of fish (control and experimental traps catch combined), with all other species classified as "other. Fish larger than 50 cm (n = 7) omitted for clearer visualization. Note different y-axes scales due to different sample sizes.
(TIF)

**S1 Table. Extended version of Table 1.** Summary of environmental and catch data for control traps (without escape gaps) and experimental traps (with escape gaps). Data for all traps and only trap pairs hauled together are presented. All means are presented ± standard deviation.
(DOC)

**S2 Table. Statistics results for differences between catch (number, mean length, biomass, number of species, and Shannon diversity index) of fish in control and experimental traps.** All tests were run on data from paired trap hauls only, with and without zero fish hauls included.
(DOC)

**S3 Table. Species with significantly higher total counts in either control or experimental trap (p<0.05 in chi-squared test of equal proportions).** Total number of fish and the percentage of the total number of fish in control or experimental traps, and chi-squared test results are shown.
(DOC)

**S4 Table. Results of Kolmogorov-Smirnov tests comparing length-frequency distributions of species in control and experimental traps.** Data are from paired trap hauls.
(DOC)

**S5 Table. Percent of fish caught in control and experimental traps less than length at maturity.** Only species for which reliable, local, length at maturity ($L_{mat}$; length at which at least 50% of individuals are mature) data are available are included. Data are from paired trap hauls. Statistic results (V-statistic and p-value) are from paired Wilcoxon signed rank tests of count data.
(DOC)

**S6 Table. Significance of fixed effects considered in the global mixed effect models.** Random effects in all models were date and individual trap ID.—p > 0.05; * p < 0.05; ** p < 0.01; *** p < 0.001.
(DOC)

**S7 Table. Linear mixed effects model summaries for global models.** Random effects in all models were date and individual trap ID.
(DOC)

**S8 Table. Linear mixed effects model summaries for most parsimonious models.** Random effects in all models were date and individual trap ID.
(DOC)

## Acknowledgments

We thank the Ministry of Agriculture, Trade, Land, Housing and the Environment in Montserrat for supporting this experiment, fishers Lester Allen, Chase Buffonge and Shane Caesar, and Dwight Sampson who was in charge of outreach. Thanks to Erin O'Reilly who created the schematic diagrams in Fig 2.

## Author Contributions

**Conceptualization:** Jason Flower, Andy Estep, Robin Ramdeen, Lennon Thomas, Sarah E. Lester.

**Data curation:** Jason Flower, Keinan James.

**Formal analysis:** Jason Flower, Claire A. Runge, Sarah E. Lester.

**Funding acquisition:** Andy Estep, Robin Ramdeen, Lennon Thomas, Sarah E. Lester.

**Investigation:** Andy Estep, Keinan James.

**Methodology:** Jason Flower, Andy Estep, Robin Ramdeen, Claire A. Runge, Lennon Thomas, Sarah E. Lester.

**Project administration:** Jason Flower, Andy Estep, Robin Ramdeen, Sarah E. Lester.

**Supervision:** Lennon Thomas, Sarah E. Lester.

**Validation:** Jason Flower.

**Visualization:** Jason Flower.

**Writing – original draft:** Jason Flower, Lennon Thomas, Sarah E. Lester.

**Writing – review & editing:** Jason Flower, Andy Estep, Keinan James, Robin Ramdeen, Claire A. Runge, Lennon Thomas, Sarah E. Lester.

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
