## [Decision Letter · Decision Letter 0]

6 Jul 2021

PONE-D-21-11222

An experimental evaluation of the effect of escape gaps on the quantity, diversity, and size of fish caught in traps in Montserrat

PLOS ONE

Dear Dr. Flower,

Thank you for submitting your manuscript to PLOS ONE. After careful consideration, we feel that it has merit but does not fully meet PLOS ONE’s publication criteria as it currently stands. Therefore, we invite you to submit a revised version of the manuscript that addresses the points raised during the review process.

Both the reviewer and myself agree that the study appears well conducted and analysed, and we have no concerns about the scientific rigour. However, there are a few points raised by the reviewer that do require clarification, particularly concerning how the fishery is characterised (as both reef fish and demersal), some clarification of the lost paired trap numbers compared to the single-hauled traps and to check the resolution and clarity of figure 6. Additionally, because PLoS One does not provide proofs, I encourage you to do a final check for any typos or other errors. My apologies for the delay in review of this manuscript.

We look forward to receiving your revised manuscript.

Kind regards,

Fraser Andrew Januchowski-Hartley, Ph.D.

Academic Editor

PLOS ONE

Journal Requirements:

2. We note that Figure 1 in your submission contain map images which may be copyrighted. All PLOS content is published under the Creative Commons Attribution License (CC BY 4.0), which means that the manuscript, images, and Supporting Information files will be freely available online, and any third party is permitted to access, download, copy, distribute, and use these materials in any way, even commercially, with proper attribution. For these reasons, we cannot publish previously copyrighted maps or satellite images created using proprietary data, such as Google software (Google Maps, Street View, and Earth). For more information, see our copyright guidelines: http://journals.plos.org/plosone/s/licenses-and-copyright.

2.1.    You may seek permission from the original copyright holder of Figure 1 to publish the content specifically under the CC BY 4.0 license. 

2.2.    If you are unable to obtain permission from the original copyright holder to publish these figures under the CC BY 4.0 license or if the copyright holder’s requirements are incompatible with the CC BY 4.0 license, please either i) remove the figure or ii) supply a replacement figure that complies with the CC BY 4.0 license. Please check copyright information on all replacement figures and update the figure caption with source information. If applicable, please specify in the figure caption text when a figure is similar but not identical to the original image and is therefore for illustrative purposes only.

Reviewers' comments:

Reviewer's Responses to Questions

**Comments to the Author**

1. Is the manuscript technically sound, and do the data support the conclusions?

Reviewer #1: Yes

2. Has the statistical analysis been performed appropriately and rigorously? 

Reviewer #1: Yes

3. Have the authors made all data underlying the findings in their manuscript fully available?

Reviewer #1: Yes

4. Is the manuscript presented in an intelligible fashion and written in standard English?

Reviewer #1: Yes

5. Review Comments to the Author

Reviewer #1: This study investigates the effect of escape gaps on trap catches in a coral reef / demersal fishery. The authors find that escape gaps had no effect on trap catch composition, size, or weight, but that traps with escape gaps had fewer catches of wide-bodied fishes (though these were rarely caught in the control trap either) and immature fishes (though this effect was weak).

The overall study result may be useful in illustrating how escape gaps can reduce bycatch, and possibly immature individuals, without impacts to fishing catch. As written, the study is clear, concise and robust. I have no concerns over the quality of the research. The findings will add to our understanding of trap catch composition and trap design. As noted by the authors, traps are a widespread coral reef fishery gear, but only very few studies on trap design. I see no issue that non-university affiliated author conducted the experimental fishing.

I have a few minor comments that may help to clarify the text.

L161 - some traps were set below 30m, presumably below limits of coral reef habitat. Throughout the text this should be clarified – that trap fishery is demersal and coral reef. This is noted in several places but the overall focus in the introduction and literature comparison was for coral reef fisheries.

L186 – how many trap pairs were hauled together vs. lost vs. hauled singly?

L201 – how many species was this?

L287 – this result seems important, if preferred food fish species are caught more in experimental trap. Could be noted in discussion – is this due to body size?

L401 – Repetition of ‘gap’ makes this sentence difficult to read, suggest rewrite

Figure 1 – blurry at full page size and difficult to distinguish trap locations. Can you use colour and make the point sizes larger?

6. PLOS authors have the option to publish the peer review history of their article (what does this mean?). If published, this will include your full peer review and any attached files.

Reviewer #1: **Yes: **James Robinson

---

## [Author Response · Author response to Decision Letter 0]

20 Nov 2021

We want to thank the editor and the reviewer, Dr Robinson, for their time and comments that have helped improve the manuscript. We are grateful for the opportunity to revise the paper.

We have addressed all the comments, including:

1. Highlighting that the fishery includes both reef and demersal species by adding clarification to the introduction and discussion.

2. Adding extra detail to the methods and results about the numbers of traps hauled and traps lost.

3. Revising Figure 1 to include colour and make the locations of the traps clearer by adding a close-up inset map. 

A detailed point-by-point response to all comments is provided in the response to reviewers document.

---

## [Editor Report · Decision Letter 1]

25 Nov 2021

An experimental evaluation of the effect of escape gaps on the quantity, diversity, and size of fish caught in traps in Montserrat

PONE-D-21-11222R1

Dear Dr. Flower,

We’re pleased to inform you that your manuscript has been judged scientifically suitable for publication and will be formally accepted for publication once it meets all outstanding technical requirements.

Kind regards,

Fraser Andrew Januchowski-Hartley, Ph.D.

Academic Editor

PLOS ONE
---

## [Editor Report · Acceptance letter]

2 Dec 2021

PONE-D-21-11222R1 

An experimental evaluation of the effect of escape gaps on the quantity, diversity, and size of fish caught in traps in Montserrat 

Dear Dr. Flower:

I'm pleased to inform you that your manuscript has been deemed suitable for publication in PLOS ONE. Congratulations! Your manuscript is now with our production department. 

Kind regards, 

on behalf of

Dr. Fraser Andrew Januchowski-Hartley 

Academic Editor

PLOS ONE